# Chimpanzees make tactical use of high elevation in territorial contexts

**Sylvain R. T. Lemoine**[1,2,3]*, **Liran Samuni**[1,2,4,5], **Catherine Crockford**[1,2,6], **Roman M. Wittig**[1,2,6]*

**1** Taï Chimpanzee Project, Centre Suisse de Recherches Scientifiques, Abidjan, Côte d'Ivoire, **2** Max Planck Institute for Evolutionary Anthropology, Leipzig, Germany, **3** Department of Archaeology, University of Cambridge, Cambridge, United Kingdom, **4** Department of Human Evolutionary Biology, Harvard University, Cambridge, Massachusetts, United States of America, **5** Cooperative Evolution Lab, German Primate Center, Göttingen, Germany, **6** Ape Social Mind Lab, Institut of Cognitive Science Marc Jeannerod, UMR5229, CNRS, Lyon, France

* sl2079@cam.ac.uk (SRTL); rwittig@isc.cnrs.fr (RMW)

## Abstract

Tactical warfare is considered a driver of the evolution of human cognition. One such tactic, considered unique to humans, is collective use of high elevation in territorial conflicts. This enables early detection of rivals and low-risk maneuvers, based on information gathered. Whether other animals use such tactics is unknown. With a unique dataset of 3 years of simultaneous behavioral and ranging data on 2 neighboring groups of western chimpanzees, from the Taï National Park, Côte d'Ivoire, we tested whether chimpanzees make decisions consistent with tactical use of topography to gain an advantage over rivals. We show that chimpanzees are more likely to use high hills when traveling to, rather than away from, the border where conflict typically takes place. Once on border hills, chimpanzees favor activities that facilitate information gathering about rivals. Upon leaving hills, movement decisions conformed with lowest risk engagement, indicating that higher elevation facilitates the detection of rivals presence or absence. Our results support the idea that elevation use facilitated rival information gathering and appropriate tactical maneuvers. Landscape use during territorial maneuvers in natural contexts suggests chimpanzees seek otherwise inaccessible information to adjust their behavior and points to the use of sophisticated cognitive abilities, commensurate with selection for cognition in species where individuals gain benefits from coordinated territorial defense. We advocate territorial contexts as a key paradigm for unpicking complex animal cognition.

## Introduction

"Be before the enemy in occupying the raised and sunny spots (. . .). Then you will be able to fight with advantage"–Sun Tzu [1]

Intergroup competition in territorial social species incurs costs to individuals, such as loss of feeding and breeding opportunities, increased risk of injuries, and death [2–5]. From an evolutionary perspective, adaptations reducing these costs and increasing competitiveness [6]

**Data Availability Statement:** All relevant data are within the paper and its Supporting Information files.

**Funding:** C.C. was supported by the European Research Council (ERC) under the European

Union's Horizon 2020 research and innovation programme (grant agreement no. 679787). L.S. was supported by the Minerva Foundation. S.L., L. S., C.C. and R.M.W. were supported by the Max Planck Society. Research at the Taï Chimpanzee Project has been funded by the Max Planck Society since 1997. The funders had no role in study design, data collection and analysis, decision to publish, or preparation of the manuscript.

**Competing interests:** The authors have declared that no competing interests exist.

**Abbreviations:** GLMM, generalized linear mixed-effects model; LRT, likelihood ratio test; TCP, Taï Chimpanzee Project; TNP, Taï National Park; UD, utilization distribution; VIF, variance inflation factor.

are expected. These adaptations can take the form of territorial tactics, minimizing the risk imposed by rivals and expressed through adaptive flexible behaviors in the face of intergroup competition. In premodern warfare involving small-scale group conflicts, humans take collective decisions to adopt risk-reduction tactics, such as ambushes and raids into the enemies' territory [7]. These low-risk tactics are adopted in conditions of numerical superiority [8], maximizing the benefits and minimizing the costs of out-group conflicts. In territorial social species, and in societies that employ flexible fission–fusion dynamics, characterized by subgroups or parties of varying sizes, conflict escalation is determined by the imbalance of power [7]. Tactical decisions to engage in a conflict at low cost, therefore, can be facilitated by early numerical assessment of traveling parties of own and neighboring groups (hereafter called "rivals") [8].

Taking coordinated decisions to adopt tactics that reduce risks involves demanding cognitive abilities [9] and the ability to adjust own's behavior based on available knowledge [10]. Thus, warfare is hypothesized to be a driver of brain size and cognition in the hominid lineage [11,12] and potentially among non-ape social species [13]. For example, warfare—a permanent but unpredictable stage of threat and regular conflicts between subgroups of varying sizes— requires tactical and coordinated decisions to outcompete out-groups, while considering the current dynamics of the threat posed by rivals. However, there are few empirical tests of this idea outside of humans [11,12], making evolutionary arguments challenging. As such, it is not known if other species use elaborated territorial tactics, nor if tactical capacities are already evident in hominids, especially in species which engage in intense territorial competition that resembles human small-scale conflicts. While assessing underlying cognition involved in territorial tactics is well recognized to be challenging [14], here we aim at taking a crucial step to assess whether under natural conditions, behavioral patterns are consistent with those expected given tactical decisions that result in minimizing risks.

To this end, we focus our behavioral analysis on the use of topography by 2 neighboring groups of wild western chimpanzees (*Pan troglodytes verus*) from the Taï National Park (TNP) [15], Côte d'Ivoire. In premodern human warfare, a common low-risk tactic favoring rivals detection and assessment of their power is the use of topography [16–18], particularly elevated areas that facilitate visual and auditory early detection of rivals [19]. Archaeological and historical evidence suggests that high elevation is used in a territorial context by humans to ambush enemies [20], to gain protection, as elevated areas are difficult to attack by enemies without detection [21], and to acquire tactical advantage over enemies through early detection of their numbers and location [22]. Whether collective decision-making [9] to use the topography to facilitate detection of rivals is a unique feature of human warfare or whether it is a shared capacity with other animals, however, has not been investigated. Use of rugged tectonic landscape has been proposed as an adequate niche during the transition between tree and ground dwelling in ancient hominins [23,24], enabling better protection against predators. Given the potential highly competitive context between conspecifics and between hominin species, controlling and exploiting key high points in the landscape could have provided significant territorial advantages. Unraveling how wild chimpanzees combine territorial behavior with landscape use would shed light on the importance of high elevated landscape throughout human evolution.

In fact, while elevation use for acoustic signaling [25] to conspecifics and for predator detection [26] is common in animals, its use in intergroup competition remains unknown, for example, whether or not information gathered at high elevation is used to gain a tactical advantage over out-groups. In territorial species where numerical assessment is key in intergroup conflicts, one would expect behaviors that (i) enhance early detection of out-groups; and (ii) enable advantageous tactical use of this information to minimize risk, such that an

advantageous situation promotes engagement with the rival groups, while a disadvantageous situation promotes retreat. In contrast to signaling and anti-predation, hostile rivals pose a higher cognitive challenge, particularly in fission–fusion societies in which groups coordinate their territorial activity and their numbers and movements regularly change.

In chimpanzees (*Pan troglodytes*), a social species with fission–fusion group dynamics like humans [27,28], and one of humans' closest living relatives, intergroup competition between rival neighboring communities likely acts as a selective pressure [3,4], favoring in-group cooperation [4,29] during territorial contexts, such as collective border patrols [30,31] and coalitionary killing of rivals [32]. As in premodern human warfare, the imbalance of power between groups determines conflict escalation [7] in chimpanzees. Chimpanzees show abilities to assess rivals' numbers from a distance by not approaching when rival vocalizations indicate numerical superiority [8,33–35]. Whether chimpanzees buffer the costs of territorial disputes through tactical use of high elevation to preemptively gather information about rivals and use this information to make tactical decisions to approach or retreat from rivals, has so far not been addressed.

Here, we hypothesized first that a tactical use of elevated areas by chimpanzees assists in early detection of rivals. We first analyzed how the likelihood to stop at the highest hills located within the overlap border area of 2 neighboring chimpanzee groups is affected by the traveling direction, own numbers, and intercommunity distances. Stop events (resting for >5 min) at high elevation offer sensory advantages, potentially allowing to gather information about rivals. However, climbing hills may be energetically costly, so that their use must be carried out when the potential benefits of rival detection outweigh energetic costs. We predicted that Taï chimpanzees would be more likely to stop at peripheral hills when traveling towards the border than towards the territory center, when in low rather than in high numbers (as risks of an unfavorable imbalance of power are higher when in low numbers), and when rivals are less likely to have already been detected—namely for longer intercommunity distances. We also considered the number of peripheral hills used before and after each occurrence of hill use per day as this may impact the likelihood to stop on a particular location. If information on rivals has already been gathered from preceding hill tops, climbing more hills could be redundant and increase energetic expenditure. Here, we predicted that, as chimpanzees move towards the territory border, they will be less likely to stop at peripheral hills if they have already stopped on previous peripheral hills.

Second, we predicted that chimpanzees preferentially engage in activities that favor information gathering about rivals (resting rather than feeding or traveling) when in elevated border areas relative to elevated central areas. Changes of activity between central area and periphery are well known in chimpanzees [33], but not whether local elevation modifies activity budgets. Consequently, to account for the impact and interplay between elevation and territory location, we tested how the interaction between elevation and location (measured by kernel values) influences the likelihood to observe chimpanzees rest, feed, and travel. In tropical forest habitat, a low-visibility environment, early detection of conspecifics occurs primarily through auditory rather than visual channels. Vocalizations and buttress drums are among the first detected long-distance signs of rivals' presence in 73% of neighboring encounters [4]. Chimpanzee pant hoots and buttress drums can be heard >1 km away [36] and elevation above the canopy level improves sound detection over long distances [37]. Thus, decisions to engage in activities (resting) that favor auditory attentiveness to distant sounds, particularly when on high elevation in dangerous border areas, could maximize acoustic detection of rivals. Activities differ in their potential to increase auditory attentiveness and in creating a suitable environment for information gathering about rivals. Feeding and traveling activities involve a particular focus on a goal and usually produce noise and often require stopping to focus on

distant vocalizations (S1 Video). Resting offers the best conditions for information gathering from the surrounding auditory environment, especially to detect distant chimpanzee sounds.

Third, we hypothesized that early detection of rivals enables tactical use of information obtained in elevated border areas to make low-risk travel decisions. Riskier situations to the in-group occur when rivals are in high numbers while own numbers are low and when rivals are located close by (<2,000 m). We used simultaneous data on 2 neighboring chimpanzee groups to assess if potential information gathering about neighboring rivals during high elevation use within the overlap border area would lead to risk-reduction tactical movement decisions to retreat from or advance toward rivals. We expected that, if chimpanzees use hills in the territory border to gain information about neighboring rivals' current numbers and location, chimpanzees' subsequent travel decisions after hill use (but not after use of low-lying peripheral areas) will maximize benefits and reduce the risks of territorial engagement. We examined the tactical movement patterns within the 30 min following each stop (>5 min) event at peripheral hills (n = 304 occurrences) and low-lying peripheral locations (n = 321 occurrences) together, with a model assessing the likelihood to advance towards rivals' location. Given that high hills may allow to gather information on both rival numbers and location, we tested in this model the effect of the three-way interaction between elevation, imbalance of power, and intercommunity distance. Fig 1 provides a graphical representation of the predictions based on this interaction. In these analyses, we assessed the imbalance of

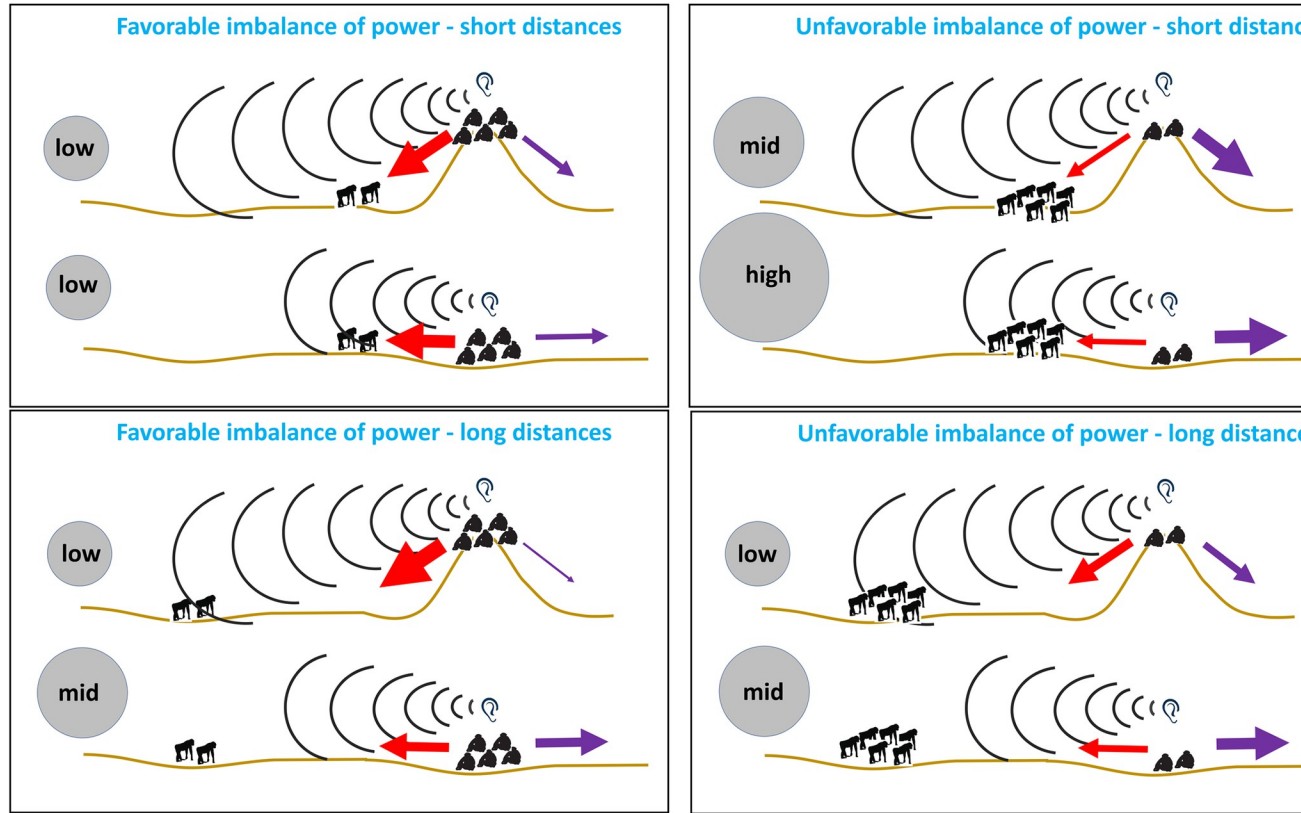

**Fig 1. Predictions for the likelihood chimpanzees advance towards or retreat from rivals' location, depending on elevation, imbalance of power, and intercommunity distance.** Sitting chimpanzee silhouettes: those gathering information (illustrated by the "ear" drawing); standing chimpanzee silhouettes: rival chimpanzees; black arc circles depict the auditory detection range that reaches further from higher elevation; gray circles depict the putative risk imposed by rivals (low, medium, high); red arrows: likelihood to advance towards rivals; purple arrows: likelihood to retreat from rivals; thickness of arrows: extent of the likelihoods.

power by the arithmetic difference between adult party size of the location users and the closest adult party size of rivals. In Taï chimpanzees, the number of adult males determines territory increase while the whole group enables an efficient territory defense [38]. Since both adult males and females are involved in intergroup encounters [39] and border patrols [31,40], the competitive ability of Taï chimpanzees is best reflected by all adults rather than by only adult males. We measured intercommunity distance at the time of departure from the specific peripheral location.

We investigated these 2 hypotheses—tactical use of elevation for information gathering and tactical decisions based on information obtained when using high elevation—in the Taï western chimpanzees (*Pan troglodytes verus*). High elevation in the Taï forest is characterized by granitic inselbergs—isolated rocky hilltops with little canopy cover—which offer ideal listening locations for early detection of rivals. We simultaneously collected behavioral and GPS ranging data between December 2013 and October 2016 on 2 neighboring chimpanzee communities who share a territory overlap area, the South and East groups [41]. Human observers conducted daily focal follows of 58 habituated individual chimpanzees for 8 to 12 h/day (10,480 and 10,706 observation hours across 1,287 and 1,350 follow-days for South and East, respectively). We continuously recorded the individuals visible within the focal party, allowing us to assess adult party sizes. We reconstructed the topographical landscape across the groups' territories using the GPS points accrued during each minute of focal sampling, recording location and elevation (range of elevation: 115 to 287 m above sea level, Figs 2, S1 and S2). Simultaneous focal follows of individuals from both neighboring communities allowed us to assess intercommunity GPS distances. Given the fission–fusion nature of chimpanzee societies, the intercommunity distance represents the minimum known distance between the neighboring groups.

## Results

### Tactical use of elevation for information gathering

**Prioritizing stop events on peripheral hills.** Within the East-South overlap area, we identified the highest hills (>230 m above sea level, corresponding to the 99th percentile of elevation range—throughout called peripheral hills) from the reconstructed landscape. We defined hill use when chimpanzees traveled within a 50 m radius of hill summits. Across the study period, 103 peripheral hills were climbed by the chimpanzees (Fig 2), 22 of which were climbed by both groups. South and East groups visited a total of 58 and 67 peripheral hills, respectively. There were 36 and 45 hills uniquely used by South and East groups, respectively.

To analyze whether chimpanzees prioritize stop events on peripheral hills, we used a generalized linear mixed-effects model (GLMM) with binomial error structure on all instances in which chimpanzees climbed a peripheral hill ($n$ = 717 occurrences). Specifically, we analyzed whether the presence (coded as 1) or absence (coded as 0) of stop events on peripheral hills (≥5 min of resting) is influenced by the previous movement direction, by the interaction between own adult party size and intercommunity distance, and by other peripheral hills use. The likelihood to stop on peripheral hills was influenced by the test predictors (full-null model comparison: likelihood ratio test (LRT), $\chi^2$ = 55.82, df = 6, $P$ < 0.001). The model confirmed our prediction (effect of traveling away from the center: estimate ± SE = 1.656 ± 0.283, $P$ < 0.001, $R^2$ = 0.087; Fig 3A and S1 and S2 Tables), with chimpanzees stopping on peripheral hills in 57.73% of the movements toward the border, but only in 25.09% of the movements toward the territory center. Stopping at a given peripheral hill was less likely when more peripheral hills were subsequently climbed that day (estimate ± SE = −0.381 ± 0.114, $P$ < 0.001, $R^2$ = 0.019; Fig 3B and S1 and S2 Tables), suggesting of conservative optimization

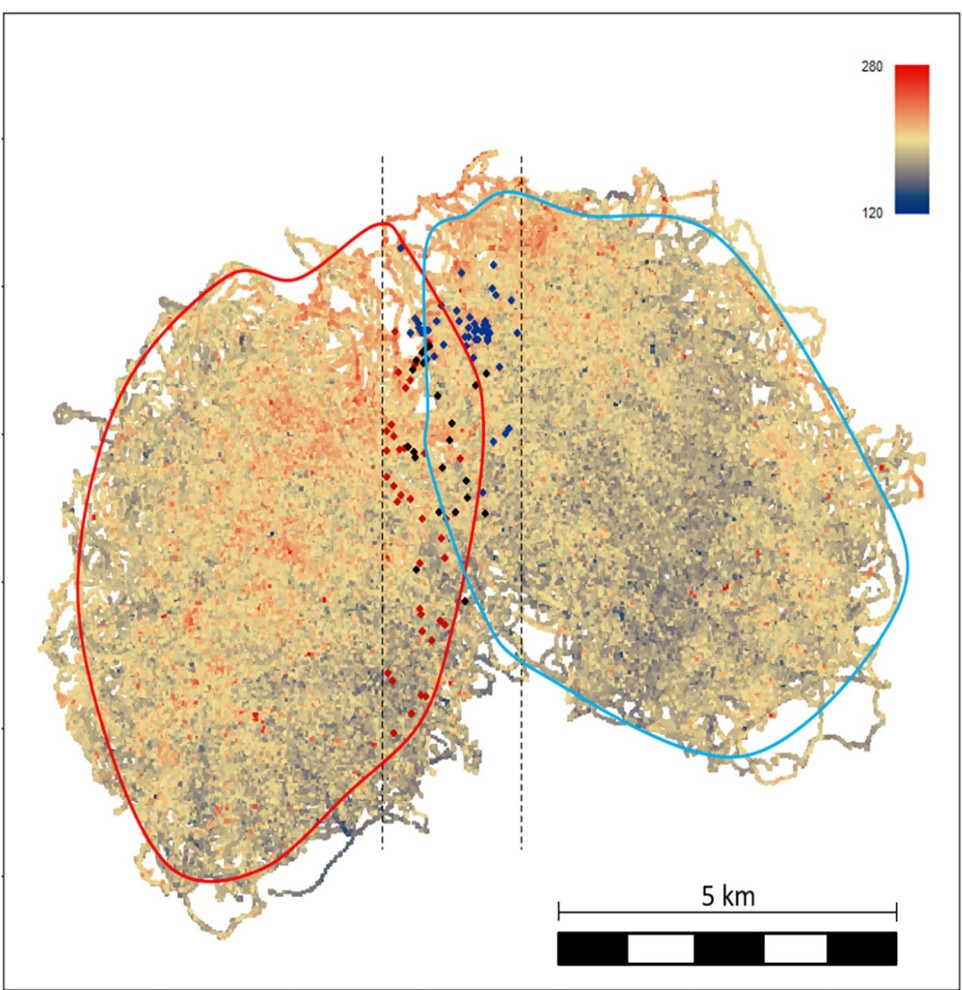

**Fig 2. Reconstructed topography.** Reconstructed topography (meters above sea level) across the territories of Taï chimpanzees South group (left) and East group (right), based on the accumulation of 21,186 h of focal chimpanzee track log follows (4,817.8 km for South, 5,168.9 km for East) between December 2013 and October 2016 from a total of 58 chimpanzees. The red line represents 95% of the kernel distribution for South group, the blue line the 95% of the kernel distribution for East group. The black dotted lines depict the peripheral area, with peripheral hills above 230 m used uniquely by South group in dark red, uniquely used by East group in blue, and used by both groups in black, during this time period. See S1 Fig for a map of the lower elevation locations.

of climbing. The number of hills previously visited had no significant effect on stop events (S1 and S2 Tables), but stop events on peripheral hills were less likely in later hours of the day (estimate ± SE = −0.302 ± 0.105, $P = 0.004$, $R^2 = 0.012$; S1 and S2 Tables). A lower likelihood to stop on peripheral hills later during that day may result from chimpanzees being more likely to return to central areas at that time, given that traveling towards the border was less likely in late hours of the day (separate GLM: estimate ± SE = −0.069 ± 0.024, df = 1, $P = 0.005$). Contrary to our prediction, stopping at peripheral hills was not influenced by own adult party size and intercommunity distance, nor by their interaction (S1 and S2 Tables).

**Prioritizing quiet activities on peripheral hills.** Resting, feeding, and traveling budgets (percentage of observation time spent in each activity) show that for both groups, the time spent resting and traveling increased with proximity to the border, while time spent feeding decreased (S3 Table), confirming previous findings on changes of activity as function of the location in the territory [33].

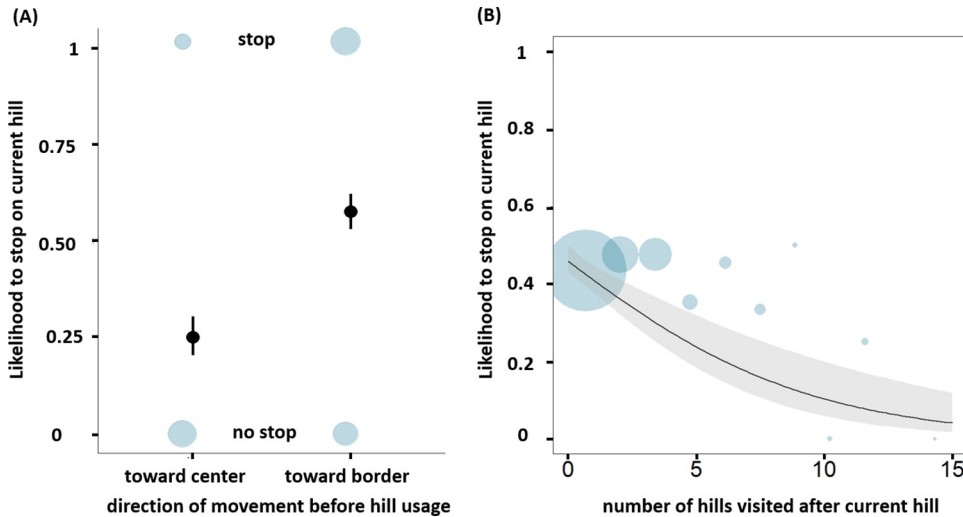

**Fig 3. Tactical use of elevation by wild chimpanzees: prioritizing stops at high elevation.** Likelihood to stop at peripheral hills as a function of the preceding movement direction (A) and as function of the number of peripheral hills visited the same day after using that particular hill (B). (A) The black dot depicts the model estimates, the black lines the 95% confidence intervals. (B) The plain line depicts the model line, the shaded area the 95% confidence intervals. (A and B) The size of the blue circles relates to the observed mean values per combination of the predictor. Likelihoods range from 0 to 1. The raw data and analyses underlying this figure may be found in S1 Data and in S1 Analysis, respectively.

To assess whether chimpanzees prioritize activities on hills that would aid information gathering and to assess potential differential in elevation use between central and border areas, we used 3 separate GLMMs with binomial error structure, one for each activity. In each model, we used a minute-point ($n = 42,385$) dataset where we coded resting, feeding, or traveling either as 1 when it was observed, or as 0 when not observed. We conducted this analysis, while accounting for temporal and spatial autocorrelation, using data from the entire territories of South and East groups. All activities were affected by the test predictors (full-null model comparisons: resting–LRT: $\chi^2 = 22.36$, df = 3, $P < 0.001$; feeding–LRT: $\chi^2 = 47.20$, df = 3, $P < 0.001$; traveling–LRT: $\chi^2 = 49.95$, df = 3, $P < 0.001$). The interaction between location and elevation significantly affected resting (estimate ± SE = 0.045 ± 0.018, $P = 0.014$, $R^2 = 0.001$; Fig 4A and S4 Table), but neither feeding (estimate ± SE = −0.017 ± 0.020, $P = 0.387$; Fig 4B and S5 Table) nor traveling (estimate ± SE = 0.014 ± 0.023, $P = 0.529$; Fig 4C and S7 Table). Resting was always more likely at high than at low elevation, the difference being more pronounced in the border area than in the central areas (Fig 4A). Importantly for our hypothesis, resting at high elevation near the border was more likely than resting at similar elevation near the center (Fig 4A). Reduced feeding and traveling models, lacking nonsignificant interactions, revealed that feeding decreased (estimate ± SE = −0.155 ± 0.017, $P < 0.001$, $R^2 = 0.005$; Fig 4D and S6 Table) and traveling increased near the border compared with the territory center (estimate ± SE = 0.111 ± 0.021, $P < 0.001$, $R^2 = 0.001$; Fig 4E and S8 Table). Traveling also decreased with elevation (estimate ± SE = −0.188 ± 0.033, $P < 0.001$, $R^2 = 0.004$; Fig 4F and S8 Table). These results are unlikely to be due to the geographic distribution of elevation, since, for both groups' entire territories, elevation near the border is significantly lower than in the territory centers (S2 and S3 Figs): periphery (kernel values above or equal to 75) [42] versus core area (kernel values below 75) average elevation per kernel value comparisons: Mann–Whitney U tests, South: W = 1699, $p < 0.001$; East: W = 1327, $p = 0.001$, S3 Fig.

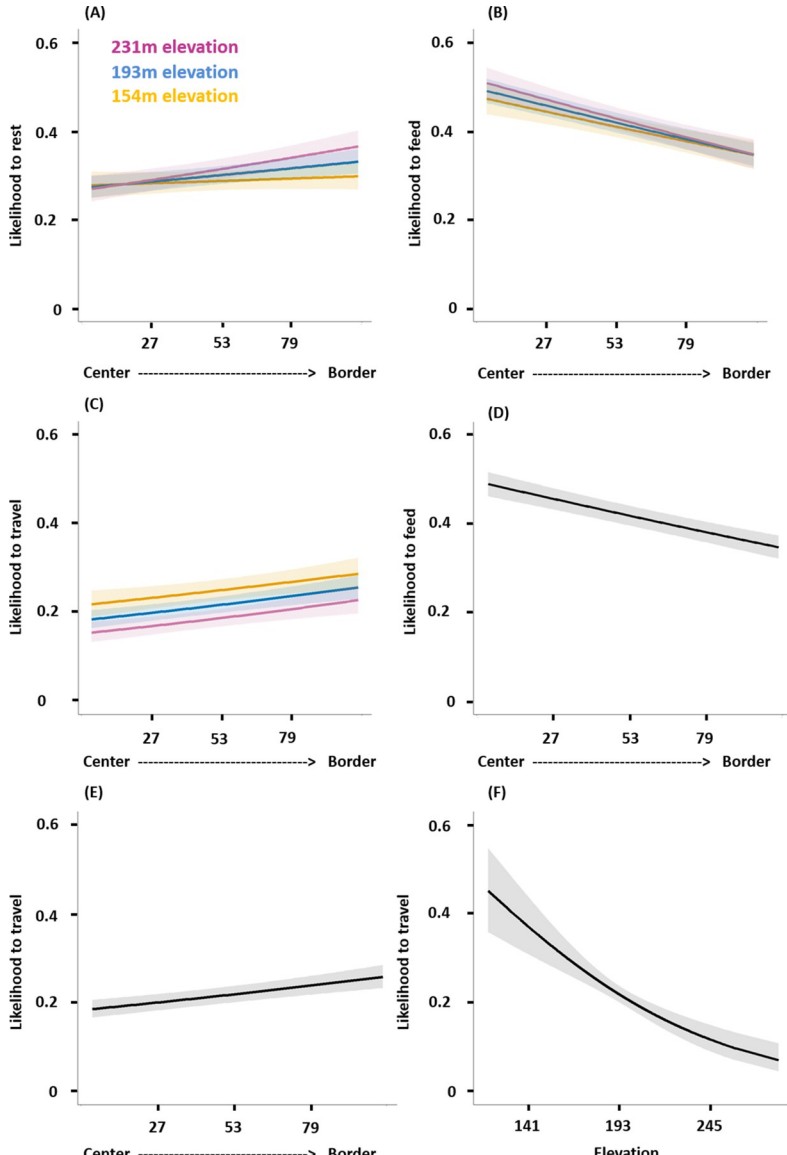

**Fig 4. Tactical use of elevation by wild chimpanzees: modulation of activities.** (A to C) Interaction between elevation and location on the likelihood to rest (A), feed (B), travel (C). Locations within the territory (x-axis) correspond to utilization distribution (UD) kernel values (4 to 99). Plain lines depict the model lines for 3 values of elevation: 154 m in orange, 193 m in blue, and 231 m in pink; shaded areas depict the 95% confidence intervals. (D to E) Effect of the location within the territory on the likelihood to feed (D) and to travel (E). (F) Effect of the elevation on the likelihood to travel. Likelihoods range from 0 to 1. The raw data and analyses underlying this figure may be found in S2 Data and S2 Analysis, respectively.

Distributions of elevation used for each activity (S4 Fig) confirmed the variation in activity budgets across the territory—that higher elevation is typically used for resting in the periphery relative to high elevation closer to the center.

## Tactical decisions based on information obtained when using high elevation

**Determinants of peripheral movements.** To examine whether elevation use informs tactical maneuvers towards rivals, we used a GLMM with binomial error structure. Specifically,

we analyzed whether elevation, imbalance of power, and intercommunity distance impact the likelihood to advance towards rivals (advance coded as 1, retreat coded as 0), after stopping at high and low peripheral locations (highest locations/hills $n = 304$, mean ± SD stopping time = 38 ± 65 min; low-lying peripheral locations $n = 321$, mean ± SD stopping time = 30 ± 32 min). To test our set of predictions, we included a three-way interaction between elevation, imbalance of power, and intercommunity distance as test predictors. We used the natural logarithm of the intercommunity distance to give more weight for shorter distances, given that rivals' detection may not vary linearly as function of intercommunity distances and may be easier and more likely at short distances. The previous movement direction of the party using the locations (toward border as 1, toward center as 0) was also tested. The likelihood to advance towards rivals after a >5 min stop at high or low locations was influenced by the test predictors (full-null model comparison: LRT, $\chi^2 = 78.59$, df = 10, $P < 0.001$). Chimpanzees were more likely to advance towards rivals during movements toward the border than toward the center (estimate ± SE = 0.748 ± 0.104, $P < 0.001$, $R^2 = 0.107$; S9 Table). The three-way interaction between elevation, imbalance of power, and intercommunity distance was not significant (estimate ± SE = 0.025 ± 0.094, $P = 0.788$, S9 Table). A reduced model lacking nonsignificant interactions revealed that the likelihood to advance towards rivals increased with a favorable imbalance of power (and thus retreat was more likely when in an unfavorable imbalance of power) (estimate ± SE = 0.258 ± 0.093, $P = 0.005$, $R^2 = 0.014$; Fig 5A and Table 1). The interaction between elevation and intercommunity distance almost reached significance (estimate ± SE = 0.187 ± 0.096, $P = 0.0537$; Fig 5B and Table 1). This interaction (Fig 5B), shows that, at short intercommunity distances (e.g., 500 m), the likelihood to advance towards rivals did not vary with elevation, while at longer distances, the likelihood to advance increased following hill top use. For example, after stopping on a peripheral hill of 250 m elevation, the likelihood to advance towards rivals increased from 40% when rivals are at 500 m to 50% when rivals are at 1,000 m and to 60% when rivals are at 3,000 m (Fig 5B). In contrast, when at lower elevations, the likelihood to advance did not change with intercommunity distance, remaining at around 40%.

## Discussion

### Summary of results and overview

We demonstrate that chimpanzees likely use high elevation to gather information about rivals and use this information to make low-risk travel decisions in dangerous parts of their territory. Chimpanzees used hill tops during border activity according to a clear set of behaviors that unfolded over several kilometers, and over a substantial number of distantly located hills, from the time of entering the periphery to returning to own territory. First, chimpanzees were more likely to climb hills (i) while traveling towards the territory border rather than towards its center; and (ii) if they had not already climbed hills during the current travel event to the territory border. Second, at high elevation, chimpanzees were more likely to engage in resting, an activity suitable for detecting distant chimpanzee-related sounds, rather than engaging in noisier activities that impede listening (feeding or traveling). A key finding is that resting was more likely at high elevation in the border than at similarly high elevation in the territory center, suggesting that resting was not related to other hill top-related factors, such as recovering from the climb or using sunny spots. This reflects a modulation of the activity as function of both the elevation and the risk inherent to the peripheral location. Third, we determined that chimpanzees use information gathered on hill tops to inform subsequent travel decisions when in border areas. Specifically, high elevation provides a tactical advantage in assessing intercommunity distances, but not rival numbers. Chimpanzees on hills, compared to low elevation,

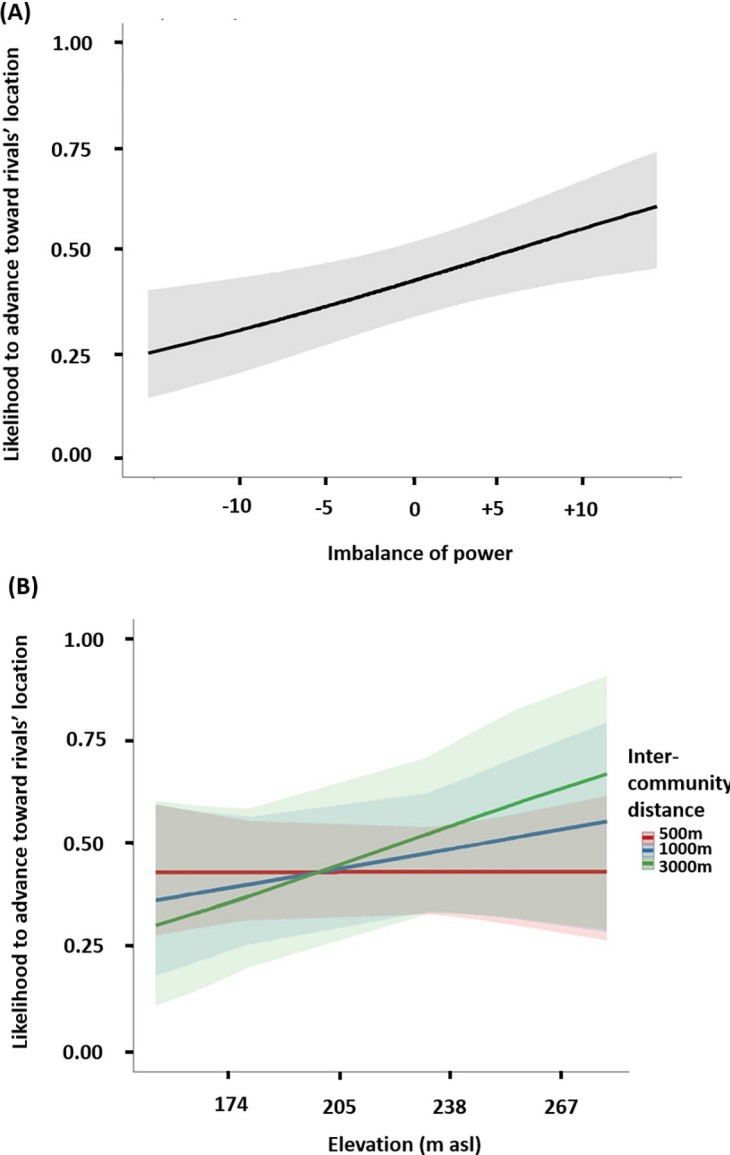

**Fig 5. Rival-related information gathered from hill tops determines movement decisions.** Likelihood to advance towards rivals' location after stopping at peripheral locations, as function of (A) the imbalance of power between own adult party size and rivals' adult party size; and (B) the interaction between elevation and intercommunity distances. The plain lines depict the model lines, the shaded area the 95% confidence intervals. The raw data and analyses underlying this figure may be found in S3 Data and S3 Analysis.

were more likely to advance when rivals were far, suggesting that hill top assessment offers chimpanzees low-risk opportunities for exploration of rival territory.

Travel decisions were consistent with the idea that information gathered on hill tops about rivals was used to minimize the risk of territorial engagement. These patterns largely fit with collective risk-reduction territorial engagement patterns well known in human territorial disputes [16,22]. Overall, our analysis of high elevation use underscores unsuspected abilities for chimpanzees to, in a low visibility environment, detect, gather, and interpret key features of rivals' situation in a competitive context and in the absence of any visual information. These

**Table 1. Determinants of the likelihood to advance towards rivals after departure from the low and high locations.**

| Terms | Estimate (SE) | z value | P-value | 95% CI |
|---|---|---|---|---|
| (Intercept) | −0.301 (0.191) | −1.570 | (i) | −0.677; 0.074 |
| Nb. Hills used before[a,b] | −0.125 (0.105) | −1.193 | 0.232 | −0.331; 0.080 |
| Nb. Hills used after[a,b] | −0.014 (0.099) | −0.148 | 0.882 | −0.209; 0.179 |
| Own movement[a,b] | 0.750 (0.103) | 7.222 | **<0.001** | 0.546; 0.953 |
| Imbalance of power[a,b] | 0.258 (0.093) | 2.776 | **0.005** | 0.076; 0.441 |
| Intercommunity distance[a,b,d] | 0.033 (0.099) | 0.337 | (i) | −0.160; 0.227 |
| Elevation[a,b] | −0.184 (0.103) | −1.781 | (i) | −0.387; 0.018 |
| Intercommunity distance * Elevation[b] | 0.187 (0.096) | 1.929 | *0.053* | −0.002; 0.377 |
| Relative distance to center[a,c] | −0.175 (0.118) | −1.486 | 0.137 | −0.408; 0.056 |
| Quadratic relative distance to center[c] | 0.074 (0.045) | 1.625 | 0.104 | −0.015; 0.164 |
| Location[a,c,e] | −0.076 (0.100) | −0.765 | 0.444 | −0.273; 0.119 |
| Time of the day[a,c,f] | −0.030 (0.095) | −0.323 | 0.746 | −0.217; 0.155 |
| Sex (males as reference)[c,g] | −0.039 (0.185) | −0.213 | 0.831 | −0.402; 0.323 |
| Group (South)[c,h] | −0.035 (0.205) | −0.172 | 0.863 | −0.438; 0.368 |
| Temporal autocorrelation term[a,c] | 0.207 (0.096) | 2.158 | **0.030** | 0.019; 0.396 |

Results of the *reduced model* lacking statistically non-significant interactions.

(a) z-transformed.

(b) Test predictors.

(c) Control predictors.

(d) Natural logarithmic transformed.

(e) Measured as kernel values from utilization distributions based on the track logs; kernel values increase with the distance to the territory center.

(f) Circadian values.

(g) Refers to males as compared to females.

(h) Refers to South group as compared to East group.

(i) Have no meaningful interpretation. Dataset $N = 625$, 2 groups (East and South), marginal effect sizes ($R^2$): 0.184 and conditional R2: 0.206. *P*-values in **bold** a statistically significant effect at $\alpha = 0.05$, *p*-values in *italic* a statistically significant effect at $\alpha = 0.10$.

results are, to our knowledge, the first evidence in nonhumans of elaborate tactical use of elevation in a territorial context resulting in subsequent low-risk travel decisions.

## Hill top use and hill top activity (when, where, and how hills are used)

Tactical use of elevation was evident in travel decisions to use peripheral hills: climbing and stopping at the highest peripheral hills was more likely during travel toward the border than toward the center. On the outward journey, chimpanzees showed a greater likelihood to stop on hills when no other hills lay ahead. These results show that: first, hills are avoidable; second, hill use appears to have an outward-bound purpose; and third, chimpanzees have knowledge of hills ahead (which cannot be seen from ground level where chimpanzees travel due to forest cover). These findings imply that chimpanzees optimize their peripheral hill usage.

Tactical use was also evident in activity decisions on peripheral hill tops. Peripheral hills were more likely to be used for resting than central hills, even though central compared to peripheral areas generally have higher elevation (Fig 2). Resting likely provides more favorable conditions for increased auditory attentiveness, in contrast to noisier activities of feeding or traveling, where individuals may also be focused on non-territorial goals. Resting at high elevation provides optimal acoustic conditions for hearing sounds that rival chimpanzees generate, such as vocalizing and drumming, which can be audible over distances of more than 1 km [36]. However, we were not able to determine whether all party members increased alertness

and vigilance while resting at high elevation near the border. Future detailed behavioral investigations, especially on coordinated behavior among party members on hill tops, would be necessary to assess these potential differences.

Taken together, the observed patterns of when and where hills are used, and the choice of hill top activity are consistent with climbing hills for the purpose of gathering information on rival group movements, which would be redundant on the return journey [10]. When in the overlap area, parties from both East and South groups travel long distances during border patrols (South: mean ± SD 2,500 m ± 2,300 m; East: mean ± SD 2,400 m ± 1,725 m). In addition, chimpanzees are known for coordinating their territorial activities across kilometers [2,27,28,40], such that chimpanzee subgroups stay together for longer duration than during other activities [31]. Consequently, restricting the use of high hills to journeys towards the border could maximize the competitive advantage over rivals while optimizing energy expenditure. While our results fit well with the idea that chimpanzees use hill tops to gather information about rivals, it remains possible that chimpanzees use hill tops for other primary purposes than rival detection, such as detecting food trees, landmark usage, or sun exposure. However, hill top vegetation is generally dense (all authors: pers. observation), and while hills offer excellent acoustic advantage they do not offer a better visual advantage to aid food or landmark detection [43]. For sun exposure, less costly travel locations, such as low elevation tree falls are available, and chimpanzees frequently use fallen trees for this purpose. A future step forward to confirm the territorial purpose of peripheral hill use could consist of expanding the link between landscape topology and usage and occurrence of border patrols, intergroup encounters, and raids.

Additionally, consistent with previous findings [44], we show that chimpanzees reduce feeding and increase traveling near the border, a modulation of activity typically seen during border patrols [31,45]. Traveling activity was also modulated as function of elevation, with less preference for higher elevated areas, consistent with an optimal traveling strategy [46–48].

## Territorial engagement following hill top use

Tactical use of high elevation to optimize low-risk territorial engagement was reflected by traveling decisions following stop events at peripheral hills. As seen previously in wild chimpanzees [34,35,40], we found that strength in number determined advances or retreat decisions, with advances being more likely (and so retreat less likely) when in situations of favorable imbalance of power. This effect, however, was not influenced by elevation usage. In contrast, high elevation appears to provide a tactical advantage in detecting rivals' location, particularly when rivals are far away. Without hilltop usage, chimpanzee vocalizations and buttress-tree drums can be clearly heard over 500 m away [49]. Hence, detecting rivals over such distances does not require hill top usage, and indeed we found limited effect of elevation at such short distances. In contrast, when rivals were further away (1,000 m), advances were about 20% more likely after hill top usage (250 m) than from low elevation (175 m), and this differential increased with intercommunity distance. This suggests that high elevation is used by Taï chimpanzees to assess the presence or absence of rivals particularly at longer distances.

Preemptive information gathering from hill tops about the rivals' location would allow to reduce the risk of subsequent travel or of an escalation of territorial conflicts in dangerous border areas. Intergroup encounters occur at border areas [28,33], thus adopting risk-reduction tactics during approaches by exploiting the landscape to their advantage can reduce the cost of being outnumbered by rivals. Our results are consistent with the possibility that in overlap range areas, chimpanzees particularly use hill tops when they have not yet heard signs of rivals.

## Cognitive capacities involved in hill top use

Knowledge about rivals' presence may be gathered while traveling toward the border, accumulating when traveling from hill top to hill top. Detection of rivals' location usually occurs through hearing rivals drumming and vocalizing, rather than by visual detection which can only occur at short range due to dense forest vegetation. Numerical estimation [50] of rivals could follow Weber–Fechner's law [51], where small numbers are easier to discriminate than large numbers. Cumulative knowledge determining numerical assessment of own power is likely less cognitively taxing [52], although ample evidence shows that chimpanzees [8,33–35] and other primates [53,54] are able to evaluate their own power relative to rivals' power. Consequently, we suggest that the consideration of own and rivals' numbers drives our results, rather than solely their own number. Further detailed observations are required to unravel detection and numerical assessment mechanisms. Also, in this study, we considered that stopping on high hills aids detection of rivals over long distances. We considered that using stop events increases the possibility to detect rivals. Whether Taï chimpanzees behave similarly in their movement decisions after climbing, but not stopping on high hills remains to be explored.

In addition to considering their own and rival numbers to inform advances or retreat decisions, chimpanzees consider the risk posed by the proximity of rivals. These behavioral patterns are congruent with those expected if chimpanzees engage in tactical decision-making in territorial contexts while they do not fit patterns expected for plausible alternatives. Hence, we infer that chimpanzees engage in tactical decision-making in this context. The observed territorial tactics enabling risk-reduction are consistent with theoretical models of economic defensibility [55,56] and with adaptations enhancing survival in a strong competitive intergroup regime [4].

While not directly tested, the use of high elevation in territorial contexts is congruent with a sequence of goal-directed, planned behaviors unfolding over time (hours) and space (kilometers). Chimpanzees are capable of intentional behavior, and they anticipate what they themselves and others can or cannot perceive and adjust their behavior depending on this perspective taking [57–61]. Under the assumption that rival detection from high elevation is not a by-product of hill use for other purposes (as discussed above), we suggest that elevation usage in this context represents a goal directed activity that unfolds over long time frames. Our findings suggest that chimpanzees fit previous knowledge of hill distribution with current activity (climbing and resting on hill tops) and future territorial goals (e.g., raiding). Such putative capacities offer the opportunity to assess the information-seeking paradigm [62]—whether primates utilize metacognition—in an ecologically valid and novel context, chimpanzee territorial behavior. In territorial contexts, metacognitive processes may assess "where" to get information (previous knowledge of hill top locations), "when" to get information (later rather than earlier hill tops), and "what" information to get (rivals' location and numbers in perspective to own numbers), as well as using information to inform decisions (minimizing the risk). Studies on information seeking show that primates gather information when required but not when redundant [10,62], which is compatible with our findings that Taï chimpanzees optimize their peripheral hill use. Given that, here, information seeking events unfold over time and space, it seems plausible that metacognitive processes fit past and current knowledge together to achieve a future goal.

Landscape heterogeneity and its adaptive usage in territorial contexts may offer a productive context for animal cognitive assessment [13] and for examining the link between outgroup conflict selective pressure and evolution of cognition [13]. Putative cognitive skills with potential to enable individuals to out-compete others and to reduce risks in intergroup

competition include, for example, spatial cognition [63] and route planning capacities [64]. Also, collective action [65] and a common goal [66] may be determinants for risk-minimization tactics, favored by strong cooperative skills inherent in chimpanzees during territorial contexts [30,31]. Integration of working, short-term and long-term memory skills [67] are likely necessary to compare information gathered on relative numerical assessment, relative location, and efficient hill use.

### Elevation usage in other species and implications for human evolution

Usage of elevated areas in animals corresponds to a variety of adaptations, such as mate attraction [68], reduction of predation risk, and thermoregulation [69–72]. While sentinel behavior in birds and some social carnivores, from high and exposed locations and directed towards the sky, mostly play an anti-predatory function [73], it also facilitates social information gathering about other group members [74]. However, using elevation to obtain information about rival groups is much less common: in meerkats, sentinel behavior, an anti-predatory behavior sometimes occurring from higher ground such as termite mounds [75], is used by males to gather information on neighboring groups before dispersal [76]. Dwarf mongooses act more as sentinels after encountering signs of rival group presence [77], showing that sentinel behavior can act in an intergroup competition context. However, whether sentinel behavior is used in anticipation of a potential conflict and if such behavior includes coordinated risk-reduction movement decisions remains unknown.

In human evolution, high ground in rugged tectonic areas has been proposed to constitute an adequate niche for early hominins during the transition from forest to savannah [23], improving opportunities for hunting of large herbivores and predation protection [24]. Our findings suggest that landscape and terrain features such as elevated highpoints may have also provided advantages over hostile rivals in intraspecific competition [78]. Consequently, under the hypothesis of an increased intraspecific competitive regime throughout human evolutionary history [78], montane and highland habitats could have constituted key strategic terrain for groups of hominins to compete, thrive, and expand.

Tactical decision-making during in-group/out-group contexts and territorial landscape usage may offer an important paradigm to gain insight into the evolution of complex socio-cognitive adaptations, particularly in relation to the evolution of hominoid and human cognition.

## Materials and methods

### Ethics statement

All research conducted as part of the Taï Chimpanzee Project (TCP) has been approved by the ethics committee of the Max Planck Society (Ethikrat der MPG 04/08/2014). All observation conducted on wild chimpanzees are non-invasive and comply to IUCN guidelines. All research conducted as part of the TCP has been approved by the Ministère de Eaux et Forêts of Côte d'Ivoire and the Office Ivorien des Parcs et Réserves.

### Study site and population

We collected data in the TCP [41], located in the TNP, Côte d'Ivoire (5˚45 N, 7˚7 W), from December 2013 to October 2016 on 2 habituated neighboring communities of western chimpanzees (*Pan troglodytes verus*): South group and East group (range number of adult males (>12 years): 5 to 6 South, 4 to 5 East; range number of adult females: 12 to 16 South, 9 to 13 East; range number of independent individuals: 33 to 39 South, 30 to 37 East). Independent

individuals are those able to travel independently from their mothers. TNP is classified as tropical lowland evergreen seasonal forest [79]. The topography is characterized as "rippled, fairly uniform but confused and crisscrossed by many highly branched streams" [80]. Elevation across the TNP varies between 100 m and 396 m [81] and across the study site between 115 and 287 m. The geological substrate is composed of migmatite, a mixture of granite and gneiss [80,82]. The plain is characterized by evergreen dense humid forest [82], and the surface is sporadically broken by several inselbergs (rocky outcrops standing out from encircling plains as more or less isolated hills), with reported maximum slopes of 25% [80]. The topography of the Taï forest is sufficiently heterogeneous [80,81], especially around areas of granite inselbergs, to enable the analysis of elevation usage by wild chimpanzees.

### Data collection

Human observers used nest-to-nest continuous focal follows on adult chimpanzees (males > 12 years old, females > 10 years old) and orphaned individuals (age range 3 to 12 years old) [83] to record party compositions and activities of the focal individual [41]. We recorded spatial data every minute using the tracklog function of GPS devices (Garmin 62 or Rino). We individually time-matched the GPS tracks with the daily observation periods. During the study period, we followed South and East groups during 855 and 847 days, respectively. Since more than 1 focal follow within a given group can occur each day, we obtained, for South, 1,287 follow-days: 637 adult females (5,147 h), 601 adult males (4,919 h), 49 orphans (414 h); for East, 1,350 follow-days: 573 adult females (4,487 h), 656 adult males (5,140 h), 121 orphans (1,079 h). The total observation time across both groups is 21,186 h. Since several focal individuals can be simultaneously followed within the same group, and therefore within the same party, we kept unique observations of unique parties, keeping observation corresponding to only 1 focal individual when several focal individuals were present within the same party. From these focal follows, we removed observations when the focal individual was out of sight, all events of hunt and intergroup encounters and we kept only adult follows. We obtained a dataset for behavioral analyses cumulating 14,057.69 h of observation (3,745.16 h South females, 3,906.84 h South males, 2,270.02 h of East females, 4,135.67 h East males). *Party sizes* and *adult party sizes* correspond, respectively, to all independent individuals and to all adult individuals present within the visibility range of the focal individual [41]. When several neighboring parties were followed simultaneously, neighboring adult party size corresponds to the aggregated number of adults from parties observed within the last 30 min. This approach assumes that individuals observed within 30 min can meet again during processes of fission–fusion and is less subject to biases due to the sampling effort. We measured *food availability* monthly on each territory, using a fruit availability index established in previous studies in this population [84], combining fruiting phenology scores (absence/presence of mature fruits), density, and mean basal areas of each tree species.

### Location measurement

We used the GPS tracks for the entire study period to calculate the kernel density estimations of the territory for both studied groups. We used the R function "kernelUD" of the package adehabitatHR (v0.4.14) [85], with a smoothing factor (h = 149) using the plug-in method, to determine utilization distributions. Each location was then assigned a kernel value (range: 1 to 99). The smaller the kernel value, the more used and central the area is. Larger kernel values represent areas closer to the border. We determined the territory overlap between the 2 communities as the area located between the extreme western location visited by East group and

the extreme eastern location visited by South group, resulting in a spatial band of 2,025 m width (West–East) and approximately 9,400 m long (North-South) (Fig 2 and S1 Fig).

## Elevation measurement

We used the elevation recorded every minute within each GPS track, from the 21,186 h of cumulated focal follows. Due to the high pace of daily observations, we obtained several measures of elevation for the same location, recorded on different days, to reconstruct a smoothened landscape. Based on previous studies using GPS devices in the Taï forest and reporting accuracy ranges of 10 m [63,64], we set up an accuracy scale of 10 m and calculated the median elevation from several tracks passing into each unique bins of 10 m by 10 m across the landscape. The variation in elevation within the same bin ranged between 8 and 13 m, so we are confident that the median of elevation within each unique bin approximates accurately the real elevation. The number of GPS points per bin varied between 1 and 5 (mean ± SD = 3.03 ± 1.17). The reference location point of the bins, from which all other bins were interpolated, was taken as the arithmetic center (mean of longitude and latitude) of all the locations. Each unique GPS location point was then included into the bin from which its distance to the bin center was the smallest, avoiding GPS locations to fall into several bins. The median elevation within each bin was allocated to the behavioral observation data. Figs 2 and S1 illustrate the reconstructed landscape. Across the territory, the average reconstructed elevation was 196 m (± SD 13 m) for South group and 190 m (± SD 12 m) for East group. Both groups lived in a similarly variable elevation landscape (range 121 to 283 m for South and 115 to 287 m for East), with the highest areas located in the central and north-eastern side of South territory, and in the north-western side of East territory (Fig 2 and S1 Fig). The distribution of elevation across each entire territory showed that the border area is lower for both groups (S2 and S3 Figs), but the overlap border area between East and South group includes relatively hilly areas of the territories. *Peripheral hills*: within the East-South territory overlap, we identified all locations above or equal to 230 m, elevation corresponding to the 99th percentile of the elevation range (range: 121 m to 287 m; mean: 193 m; SD: 13 m), so this cut off elevation targets the highest hills. All locations equal or above 230 m being located within 50 m range were considered as a unique peripheral hill, to capture the potential connections between high elevated areas due to hill ridges. Mean (± SD) of the distances between peripheral hills was 634 m ± 431 m (range: 50 m to 1,962 m). We identified a total of 103 peripheral hills above 230 m (Fig 2), among which 22 were used in common by both groups. South group and East group used, respectively, a total of 58 and 67 peripheral hills. Across both groups and the study period, each hill was used on average 5.73 times (median ± SD: 5 ± 5.08; min: 1; max: 30). South group used each hill on average 5.98 times (median ± SD: 5 ± 4.48; min: 1; max: 20), while East group used each hill on average 5.52 times (median ± SD: 3 ± 5.58; min: 1; max: 30). *Lower elevation locations*: within the East-South territory overlap, we identified the locations where either South or East group stopped their traveling movement for resting (pause of minimum 5 min) and for which the elevation was maximum 180 m (corresponding to the mean elevation minus 1 SD). We identified a total of 251 lower elevation locations (S1 Fig). South group and East group used, respectively, 176 and 75 locations. Each lower location was used to stop and rest on average 1.28 times (median ± SD: 1 ± 0.72; min: 1; max: 6) for both groups, 1.33 times (median ± SD: 1 ± 0.77; min: 1; max: 6) for South and 1.16 times (median ± SD: 1 ± 0.57; min: 1; max: 5) for East.

## Activities

We computed the recorded activities of the focal individual for each minute of observation. We focused on 3 main activities aimed to encompass the main observed daily behavior:

feeding, resting, and traveling. Feeding and resting activities could take place on the ground or in trees, while traveling activity included only terrestrial movements.

## Intercommunity distance

We calculated the distances between one's own party and rivals party by using their GPS locations. For each usage of a particular location (peripheral hill and low elevation locations), we considered the average distance during the total duration of usage of the location (between the time of arrival to the location and the time of departure from the location), except for the models on advances where we used the distance at the time of departure from the location. When several independent and spatially segregated parties within a rival group were followed, we considered the distance of the closest party to the focal group using a peripheral hill/lower location. It is important to note that in Taï chimpanzees, communities are considerably more cohesive than in other populations such that adults spent less than 5% of their time alone, and between 45% and 65% of their time in mixed sex parties [86] or within 1 km of each other [87]. However, although very unlikely, this method does not guarantee that any undetected party would have been located in between the 2 observed ones.

## Movement/Advance/Retreat variables

We defined all occurrences of usage of peripheral hills and low elevation locations whenever a followed individual/party traveled within 50 m radius of the location, given a similar used elevation than the location center. For each occurrence of usage of the peripheral hills, we extracted whether the focal individual/party stopped (pause for minimum 5 min) or not at that location (*likelihood to stop*), and whether the traveling movement before passing by the hill was directed toward the periphery or toward the center of the territory (direction of movement). When the movement direction was unclear (following an axis north-south for example), we did not consider this occurrence of hill/low location usage. Whenever a stop event occurred at peripheral hills, and for each usage of low elevation location, we calculated whether, after using the hill/lower location, the focal individual/party advanced towards or retreated from the location of closest rival party, location at the moment of the hill/low elevation location departure (*likelihood to advance toward rivals*). Departure from the peripheral location was considered when the followed individual/party traveled away from the location for more than 50 m. We also extracted the number of peripheral hills (only within the overlap area) used the same day *before and after* the usage of each particular peripheral hill/lower location.

## Statistical analysis

**Activity analyses.** We analyzed how the likelihood to observe certain activities varies as function of the elevation and the location within the entire territory. Since observations taken each minute are strongly dependent, leading to a high spatial and temporal autocorrelation, we reduced autocorrelation issues by, first, selecting, for each day of observation, data points spaced by 30 min [88]. We chose 30 min intervals as it corresponds to twice the average lengths of activity bouts (resting: East group 11.6 min, South 15.07 min; feeding: East 16.46 min, South 17.66 min; traveling: East 5.61 min, South 7.15 min). We obtained a reduced dataset of 42,385 minute-points. Second, to assess the extent of spatial and temporal autocorrelation, for each model, we calculated and reported the average of the correlation coefficients between the differences in residuals issued from the full models and the differences in time between sampling points (temporal correlation), and the average of the correlation coefficients between the differences in residuals issued from the full models and the metric distances between sampling

points (spatial correlation). For each activity model, spatial correlations never exceeded 0.01 and temporal average correlation coefficients were relatively low (max 0.045), so we are confident that spatial and temporal correlations were limited in all our activity models. We used GLMMs with binomial error structure and a logit link function. For each minute-point, we coded each activity (resting, feeding, and traveling) either as 1 when it was observed, or as 0 if it was not observed. Each binary-coded activity was set as the response in 3 separate models, for each activity (resting: S4 Table; feeding: S5 Table; traveling: S7 Table). We included the interaction between elevation and location (measured with utilization distributions UD kernel values) as main test predictor. We included the sex of the focal individual (coded into 3 categories: male, female, and oestrus female when the focal individual was a female presenting exaggerated sexual swellings) and the groups' identities as control variables. Activity budgets are known to vary depending on the season [89], so we included the sine and cosine of the days as control variables. Activities also depend on socio-ecological factors [90,91], so we controlled for the party size, monthly food availability, and number of females presenting exaggerated sexual swellings (those present in the party with the focal individual during each observation point, not including the focal if she was presenting exaggerated sexual swellings). We included the combination of date and group and identity of the focal individual as 2 separate random effects. Temporal and spatial average correlation coefficients were, respectively, 0.017 and 0.0032 (*full resting* model); 0.044 and 0.0008 (*full feeding* model); 0.045 and 0.0010 (*reduced feeding* model lacking the nonsignificant interaction); 0.025 and −0.0003 (*full traveling* model); 0.025 and −0.0003 (*reduced traveling* model lacking the nonsignificant interaction).

## Peripheral hills analyses

After identifying the highest locations (>230 m) within the East-South overlap area, we identified all occurrences of a party climbing these hills across the study period ($n$ = 717 occurrences). We used GLMM with binomial error structure and a logit link function to analyze the determinants of the likelihood to stop at peripheral hills, coded either as 1 when a stop event occurred (>5-min pause), or as 0 for no stop event. A 5-min cut off (about a third of all resting bouts lengths) is chosen to reduce events in which chimpanzees stop briefly to wait for conspecifics and to maximize events when GPS locations match the behavior (resting), given that the GPS device is carried by a human observer who has to climb up hill with some distance behind the chimpanzees. A 5-min cut off is a reasonable period to detect potential distant vocalizations, given that chimpanzees do not tend to vocalize or drum often. In this "stop" model, we included as test predictors the following variables: direction of the movement before arrival at the hill (toward the border or toward the center, as a categorical predictor), number of peripheral hills previously passed by that day, number of peripheral hills passed by after that particular hill was used, adult party size of the followed group, and intercommunity distance. We included the interaction between adult party size and intercommunity distance (S1 Table).

To analyze movements from peripheral locations, we built a dataset including the stopping events at peripheral hills ($n$ = 304 occurrences) and stopping events to rest at peripheral low locations (<180 m) ($n$ = 321 occurrences), within the East-South overlap area. We used a GLMM with binomial error structure to model whether, after stopping at a peripheral location, subsequent movement was directed toward the rivals' location or not (advance toward rivals, coded binary 0/1, S9 Table). We included as test predictors the following variables: direction of the movement before arrival at the hill (toward border as 1, toward center as 0—a continuous variable allowed a lower model complexity than a categorical predictor), number of peripheral hills previously used that day, number of peripheral hills used afterward, elevation, imbalance of power (measured by the arithmetic difference between own adult party size

and rivals' adult party size), and intercommunity distance. Given that detection of rivals may vary nonlinearly as function of the intercommunity distance (given that detection is expected to be more likely at shorter distances than at longer ones), we used the natural logarithm of the intercommunity distance variable, before z transformation. Natural logarithm allows to give higher weight for smaller values than for larger ones. Since variation in the responses may be multi-factorial, we included the three-way interaction between elevation, imbalance of power, and intercommunity distance.

All models ("stop," "advance toward rivals") included as control variables the relative distance of the used location to the territory center (measured by the absolute distance between the location and the territory center divided by the yearly territory size of the group passing over the location), the quadratic term of the relative distance to the territory center, the UD kernel value of the location, its elevation (except for the models on advances, where elevation was set as a test predictor), the time of the day at arrival at the location, the sex of the focal individual (male or female), and its group ID. We included the random intercept of the location ID, of the dates nested in groups, and of the focal individual. Random slopes were kept minimal to variables showing nonzero variances to reduce model complexity [92].

We estimated the extent of temporal and spatial autocorrelation using the same methods as above. Temporal and spatial average correlation coefficients were respectively 0.101 and 0.007 (full model *likelihood to stop*), 0.100 and 0.007 (reduced model *likelihood to stop*), 0.224 and 0.012 (full model *advance toward rivals from high and low locations*), 0.226 and 0.013 (reduced model *advance toward rivals from high and low locations*). Since temporal correlations were relatively high in these models, we included a temporal autocorrelation term as an additional fixed effect into the models. We calculated a temporal autocorrelation term for each data point by averaging the residuals of all other data points of the same group, with the contribution of the residuals being weighted by their time lag to the particular data point. The weighting function followed a normal distribution with its standard deviation determined by maximizing the likelihood of the model with the temporal autocorrelation term included.

All statistical analyses were performed in R 3.6.1, using the functions *lmer* or *glmer* of the R package "lme4" (v 7.3–51.4) [93]. We used likelihood ratio test (R function *anova* with argument test set to "Chisq" [94]) to establish the significance of the full as compared to the null models (comprising only control predictors, random effects and slopes). To allow for a likelihood ratio test, we fitted the models using Maximum Likelihood rather than Restricted Maximum Likelihood [94]. When an interaction term had no significant effect, we ran reduced models lacking these nonsignificant interactions. *P*-values for the individual effects were based on likelihood ratio tests comparing the full with respective reduced models (R function *drop1* [95]). We checked for overdispersion and leverage issues for the binomial models. Dispersion statistics are provided under each respective model table. We checked for model stability by excluding each level of the random effect one at a time from the data and comparing the model estimates derived from these subsets of the data with those derived from the full dataset. This revealed that our models were stable. All continuous variables were z-transformed to a mean of zero and standard deviation of one. We derived confidence intervals from the models using parametric bootstraps (function *bootMer* of the "lme4" package). We assessed collinearity by deriving variance inflation factors (VIFs) [96] with the function *vif* of the R package "car" [97] applied to a standard linear model lacking the random effects. Collinearity was not an issue in any model (reported under each respective model table). We calculated models' effect sizes ($R2$) with the function *r.squaredGLMM* from the R package "MuMin" and the function *partR2* from the package "partR2." For each model, we reported the variance explained by the fixed (marginal- $R^2$) and fixed and random (conditional- $R^2$) effects for the whole model. Marginal effect sizes for significant predictors are provided in the main text. All

tests were two-tailed, and findings were considered as significant at $p < 0.05$ and as a trend at $p < 0.10$.

## Supporting information

**S1 Table. Determinants of stopping events at peripheral hills.** Results of the *full model* including the interaction between one's own adult party size and intercommunity distance. (DOCX)

**S2 Table. Determinants of stopping events at peripheral hills.** Results of a *reduced model*, lacking statistically nonsignificant interactions. (DOCX)

**S3 Table. Activity budgets (percentage of observation time) across 4 sections of the territories (core area, kernel <25; post-core area, kernel 25 to 50; pre-periphery, kernel 50 to 75; periphery, kernel >75) for both groups and per group.** Percentages are calculated within each territorial section. (DOCX)

**S4 Table. The effect of the location and elevation within the whole territory on chimpanzee *resting* activity.** Results of the *full model* including the interaction between location and elevation. (DOCX)

**S5 Table. The effect of the territorial location and elevation on chimpanzee *feeding* activity.** Results of the *full model* including the interaction between location and elevation. (DOCX)

**S6 Table. The effect of the territorial location and elevation on chimpanzee *feeding* activity.** Results of the *reduced model* lacking the nonsignificant interaction. (DOCX)

**S7 Table. The effect of the territorial location and elevation on chimpanzee *traveling* activity.** Results of the *full model* including the interaction between location and elevation. (DOCX)

**S8 Table. The effect of the territorial location and elevation on chimpanzee *traveling* activity.** Results of the *reduced model* lacking the nonsignificant interaction. (DOCX)

**S9 Table. Determinants of the probability to advance towards rivals' location after departure from the sampled locations (low and high elevation).** Results of the *full model* including the interactions. (DOCX)

**S1 Fig. Spatial distribution of the low elevation locations (<180 m asl) within the reconstructed topography (meters above sea level) of the territories of South group (left) and East group (right).** The red line represents 95% of the kernel distribution for South group, the blue line the 95% of the kernel distribution for East group. The black lines depict the peripheral area, with lower locations below 180 m used by South group in dark red and used by East group in blue. (DOCX)

**S2 Fig. Distribution of average (red), minimum (green), and maximum (blue) elevation (m asl: meters above sea level) as function of the location in the territory, for South (A) and

East (B) groups.
(DOCX)

**S3 Fig. Comparison of the average elevation per kernel value between the core area (kernel values below 75) and the periphery (kernel values above or equal to 75), across the entire territories of South (A) and East (B) groups.**
(DOCX)

**S4 Fig. Distributions of the used elevation, across the whole territory, for each activity (resting, feeding, traveling), for both South and East group together.** Distributions are split between 4 territorial categories: core area (kernel values <25), post-core area (kernel values 25–50), pre-periphery (kernel values 50–75), periphery (kernel values >75).
(DOCX)

**S1 Video. Illustration of Taï chimpanzees stopping their movement after hearing in-group conspecific vocalizations to increase attentiveness and auditive capacities.**
(AVI)

**S1 Data. Raw data underlying the analysis investigating the likelihood to stop at peripheral hills.**
(XLSX)

**S2 Data. Raw data underlying the analysis investigating the modulation of activities as function of location and elevation.**
(XLSX)

**S3 Data. Raw data underlying the analysis investigating the likelihood to advance toward rivals after stopping at peripheral locations.**
(XLSX)

**S1 Analysis. Script of the analysis investigating the likelihood to stop at peripheral hills.**
(TXT)

**S2 Analysis. Script of the analysis investigating the modulation of activities as function of location and elevation.**
(TXT)

**S3 Analysis. Script of the analysis investigating the likelihood to advance toward rivals after stopping at peripheral locations.**
(TXT)

## Acknowledgments

We thank the Ministère de l'Enseignement Supérieur et de la Recherche Scientifique of Côte d'Ivoire, the Ministère de Eaux et Forêts of Côte d'Ivoire, and the Office Ivorien des Parcs et Réserves for permissions to conduct the study. We are grateful to Christophe Boesch, who founded the field site, the Centre Suisse de Recherches Scientifiques en Côte d'Ivoire and the staff members of the Taï Chimpanzee Project for their long-term support. The Max Planck Society provides core funding for the Taï Chimpanzee Project since 1997.

## Author Contributions

**Conceptualization:** Sylvain R. T. Lemoine, Catherine Crockford, Roman M. Wittig.

**Formal analysis:** Sylvain R. T. Lemoine.

**Funding acquisition:** Catherine Crockford, Roman M. Wittig.

**Investigation:** Sylvain R. T. Lemoine.

**Methodology:** Sylvain R. T. Lemoine, Liran Samuni.

**Project administration:** Liran Samuni, Catherine Crockford, Roman M. Wittig.

**Resources:** Catherine Crockford, Roman M. Wittig.

**Supervision:** Catherine Crockford, Roman M. Wittig.

**Validation:** Sylvain R. T. Lemoine, Liran Samuni, Catherine Crockford, Roman M. Wittig.

**Writing – original draft:** Sylvain R. T. Lemoine.

**Writing – review & editing:** Sylvain R. T. Lemoine, Liran Samuni, Catherine Crockford, Roman M. Wittig.

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
