## [Editor Report · Decision Letter 0]

4 Oct 2022

Dear Dr Lemoine, 

Thank you for submitting your manuscript entitled "Tactical use of high elevation in territorial contexts in chimpanzees" for consideration as a Research Article by PLOS Biology.

Your manuscript has now been evaluated by the PLOS Biology editorial staff, and I am writing to let you know that we would like to send your submission out for external peer review. I should warn you that as we were unable to secure advice from an external expert, we're not wholly persuaded of the advance over the existing literature, so we will be looking for enthusiastic comments from the reviewers.

Once your full submission is complete, your paper will undergo a series of checks in preparation for peer review. After your manuscript has passed the checks it will be sent out for review. To provide the metadata for your submission, please Login to Editorial Manager (https://www.editorialmanager.com/pbiology) within two working days, i.e. by Oct 06 2022 11:59PM.

Kind regards,

Roli Roberts

Roland Roberts, PhD

Senior Editor

PLOS Biology

rroberts@plos.org

---

## [Decision Letter · Decision Letter 1]

9 Jan 2023

Dear Dr Lemoine,

Thank you for your patience while your manuscript "Tactical use of high elevation in territorial contexts in chimpanzees." was peer-reviewed at PLOS Biology. It has now been evaluated by the PLOS Biology editors, an Academic Editor with relevant expertise, and by four independent reviewers. Please accept my additional apologies for the delay incurred over the busy holiday period.

In light of the reviews, which you will find at the end of this email, we would like to invite you to revise the work to thoroughly address the reviewers' reports.

As you will see below, the reviewers are broadly positive, but each raises a number of concerns that will need to be addressed before further consideration. For example, you’ll see that reviewer #1 wants you to clarify several aspects of their approach, and is uncertain whether you can show that the behaviour at the top of hills is specific to that location. Reviewer #2 is very positive, but questions one of your claims, asks you to consider the meta-cognition literature, and wonders whether the two groups of animals were consistent in their behaviour. Reviewer #3 is also positive, but wonders about two possible alternative explanations (scouting for routes, time-of-day effects); s/he also questions some unexpected behaviour, and asks for several clarifications. Reviewer #4, like the others, is broadly positive, but has difficulties with the density of the manuscript, which makes it hard to follow what the you actually did (several of these concerns, very clearly laid out, overlap with those raised by the other reviewers); this reviewer also raises problems with your presumptive use of words like “tactical.”

Given the extent of revision needed, we cannot make a decision about publication until we have seen the revised manuscript and your response to the reviewers' comments. Your revised manuscript is likely to be sent for further evaluation by all or a subset of the reviewers.

**IMPORTANT - SUBMITTING YOUR REVISION**

*Re-submission Checklist*

*Published Peer Review*

*PLOS Data Policy*

*Blot and Gel Data Policy*

Sincerely,

Roli Roberts

Roland Roberts PhD

Senior Editor

PLOS Biology

rroberts@plos.org

REVIEWERS' COMMENTS:

Reviewer #1:

This is an impressive data set from two neighbouring communities of wild chimpanzees that has been thoughtfully analysed to address interesting questions regarding the use of high elevation by chimpanzees in border areas to gather information on rivals that can be used to minimise the risk of disadvantageous inter-community encounters.

I have a number of suggestions which focus on making the paper easier to read and follow for the reader and a few analysis queries. There were a few places where I didn't understand the approach that was being taken to address a question, that raised concerns that were later allayed when details of all parts of the analysis were provided - please try and ensure you give the reader an accurate overview of how you are going to tackle your questions to prevent other readers having this experience!

L82 Spelling - assess not asses

L96-108- I did not find this a good summary of the model run to test this prediction. Before I had read the results I commented "I don't understand the rationale behind this comparison. It is well known that chimpanzes behave cryptically in border areas (less likely to feed, frequent stops/rests to listen) - so surely this is what you would expect comparing behaviour from a central to a periphery area regardless of elevation. I would like to see an elevated duration of resting in high vs low elevation locations in border areas to be convinced that they are resting to listen / monitor the neighbours". Please try and make it clearer that you test for an interaction between elevation and border/central location to test your prediction

L171-80 - unusual to summarise the results in the intro - I would remove this unless it's a journal requirement

L254: Hill top information determines advances towards rivals

In this model you only include occasions where the chimps stopped on the hills. How do you know the behaviour is specific to having stopped on the hills unless you compare it to instances when they do not stop on hills

I have now read on and seen you run a separate model for stops on high hills and low hills. I am confused as to why you did not run a single model with these 2 data sets to test for an interaction between stop elevation and rival information. I think this is what you need to show that rival information more likely influenced decisions when stops were on high elevations. 

L335-337 / 387-388 - The summaries of this finding matches the intro, but doesn't seem to capture the crucial interaction in the model that showed that rest was more likely at high elevation in border areas rather than central high areas or low border areas.

L364 - you don't present data on intentional use of this information so I would remove this claim

Discussion - I think this could be improved by shortening and focussing on interpretation of the results. There are quite a few more speculative paragraphs (e.g. on social cognition and selection pressures) that I think could be removed, and careful editing to remove repetition and more tangential points would really improve this. Some parts read as if they have been inserted to satisfy reviewer critiques, but often the potential issue they counteract is not very well explained, making the relevance of these parts less clear and slightly random to the reader. 

L616 - of more than 50m - should be for more than 50m

Reviewer #2:

The authors present evidence of tactical use of high elevation locations by chimpanzees at the periphery of the territory. The authors argue that higher elevations allow for gathering better auditory information about neighbouring (rival) chimpanzee groups and that chimpanzees use to their advantage in intergroup competition (warfare). The authors use behavioural and environmental data of two neighbouring chimpanzee groups in the Taï National Park, Côte d'Ivoire, across several years. Their main arguments for tactical use of high elevations are that i) chimpanzees are more likely to rest at higher elevations (allowing for information gathering) on the way towards a potential intergroup encounter ii) that chimpanzees do use information gathered about the neighbouring group (e.g. size, distance) to make tactically sensible decisions about their own iii) that they do not show these responses to information about neighbouring groups after resting periods at lower elevations. 

I think this is a great, and unique addition to our understanding of intergroup interactions. We get further insight into chimpanzee decision making and, as the authors argue, it also has implications for understanding the role of conflict in hominin evolution. Overall, the analysis seems sound (with the caveat that I am not an expert in using GPS data) and the conclusions supported by the results. There are some slight inconsistencies in how strongly the authors interpret the tactical use of higher elevations (i.e. that chimpanzees move to a higher elevation intentionally with the purpose of getting information on neighbouring groups) but I think these are relatively minor. The majority of my comments are suggestions are cosmetic rather than fundamental to the manuscript and are listed below: 

Interpreting intentionality of elevation use

The authors very accurately state their aims to empirically test for "behavioural patterns are consistent with those expected given tactical decisions that result in minimizing risks" (L53-54). The evidence presented does support this claim, which is a significant addition to the literature. There is good evidence that they are responding to information about the rival group. I think it is more difficult to argue for the intention of spending time at elevation to gather information on rival group activities with the data presented here and the general complexity of observational data in natural environments where we know there are a large number of factors that are likely to involved in individual decision making at any one point. The authors do briefly acknowledge that there are potentially alternative explanations for resting pattern observed (Line 379). However, this feels imbalanced in comparison to the author's argument that "it is likely that hill climbing for information-gathering is done intentionally" (Line 384) and another similarly strong statement on line 485. 

The cognitive abilities related to tactical use of elevation

The authors suggest that the ability to take the perspective of what others can see or hear demonstrated in behavioural experiments supports their claim that chimpanzees could be using the elevation strategically. Could I suggest that the metacognition, especially information-seeking, literature would be more appropriate for interpreting the behaviour being studied here? It aims to understand how animals modify their search behaviour in response to their own knowledge (or lack thereof) as well as the quality of potential information sources. I think the current results would also be of a lot of interest to researchers in this area. I have suggested a short review that summarises the evidence for information seeking and highlights key papers: 

Marsh, H. L. (2019). The information-seeking paradigm: Moving beyond 'if and when' to 'what, where, and how.' Animal Behavior and Cognition, 6(4), 329-334. https://doi.org/10.26451/abc.06.04.11.2019

The last section of the discussion (L494-507) lists a number of areas of cognition that could be involved in decision-making in this context. To avoid it just being a list, it would be good to have some expansion on specifically how using higher elevation could be used to study these areas of cognition. 

Group differences 

Out of interest, were the patterns of decisions consistent between the two groups studied? I see from the models that this was not formally tested, that group membership was a control. I assume this was to reduce model complexity - given that there were already 4-way interactions involved - but I am curious as to whether the strategy was group specific or observed in at least these two groups. 

Minor Points

Line 36-38 This sentence is quite difficult to parse. Maybe it could be re-ordered or re-worded to make it easier to interpret. There are a few other very long, complex sentences with multiple subclauses dotted throughout the manuscript that could also benefit from simplification. 

Line 45-47 See previous point

Line 110 Word missing: "Riskier situations to the in-group"

Line 124 The clarity of Table 1 could be improved. Some suggestions: only have "Risk minimization movements" once; remove the parentheses, change to text for quicker interpretation ('advancing increases risk', instead of just "increased risk".

Line 174-180 Could this sentence be broken up?

Line 384 The authors suggest there are multiple studies supporting their statement but only cite one

Line 412-414 Could the authors expand on the reasoning here. It's not clear exactly how this fits together. 

Line 453-454 I think this is very interesting, these contexts could provide a stronger test of the intentional information seeking hypothesis. 

Supplementary Information, S1A Table

Typo 1: Variable given different names as a main and interaction effect ("own party size" and "adult party size")

Typo 2: "(b) test predictors, in bold" - test predictors not in bold

Reviewer #3:

In this manuscript, the authors investigate an exciting, creative, and - to my knowledge - novel question: do humans' closest chimpanzee relatives exhibit tactical use of high elevation for information-gathering in intergroup contexts? This work is likely to be of high interest, with relevance to behavioral ecology, human evolution, and psychology. It is also possible only because the authors have access to a unique field site where they have habituated, and collect concurrent data on, multiple neighboring chimpanzee communities. The authors address their question with a number of thoughtful, well-reasoned analyses. I am very much in support of publishing this work and offer several points for the authors' revision.

The body of results are consistent with the authors' interpretation but two additional alternative explanations - at least for visitation of hilltops - come to mind: 

(1) when traveling outside of their familiar core area, chimpanzees may seek vantage points to determine the appropriate route (and/or identify feeding locations), not just to survey rivals. Such scouting may not be necessary when returning to core areas. 

(2) The tendency to visit/rest at hilltops may be influenced by the time of day, with animals tending to move more quickly and efficiently (avoiding hilltops and/or rest) at later times of day when they are returning to their nesting site before dark (and/or because it has already become dark and little vantage is offered by hilltops), than at earlier hours of day when they have time for leisure and/or better conditions for perception. Chimpanzees might have fewer stop events on peripheral hills on their return to the territory center because these occur later in the day, when visibility is worse and/or when chimpanzees are motivated to more quickly get back to their nesting site.

If the authors have data on movement out of core areas to non-peripheral areas (e.g., foraging trips), it would be possible to compare visitation of hilltops during foraging versus patrol events, controlling for time of day. The tactical elevation hypothesis predicts that the direction of travel should determine visitation of hilltops, independent of time of day. If tactical use of elevation is specific to intergroup monitoring, it should be observed more on journeys to patrol the periphery than on journeys to feed in non-peripheral, non-core areas. In the event that such metrics are not easily available and the authors are unable to pursue this additional analysis, I would still advocate the paper's publication. However, if the authors cannot exclude these alternative explanations with additional analyses, the explanations should be discussed in the text. The existing finding that time of day is related to hilltop visitations certainly warrants more discussion.

In my view, the weakest findings (or those that conform least clearly to the authors' hypotheses and interpretations) are presented in lines 255-278. One set of findings is predicted and the other is not: When rivals are retreating or staying put, chimpanzees respond in ways that are consistent with potentially comparing party numbers (advancing when own number are higher and when rival numbers are lower). However, when rivals are advancing, chimpanzees respond in unexpected ways (advancing when own numbers are lower and when rival numbers are higher). If chimpanzees did not detect their rivals during rivals' (potentially quiet) approaches, as the authors suggest, there should be no effect of own number or rival number. Inspection of the graphs suggests that this might actually be the case, at least for the own party number finding. It would be useful to see more discussion of this, and of any reasons why chimpanzees might be more likely to approach larger groups of rivals. Is this somehow an artifact of the models/interactions? Given the slightly mixed support for information-gathering in this section, it may also be worth changing the header to be more neutral (i.e., to not imply a claim of information-gathering). That said, I do acknowledge that the bulk of the results are in line with information-gathering, and that this body of findings is informative and interesting and add value to the manuscript.

Because chimpanzees live in fission fusion societies, at any given time a chimpanzee community may be comprised of a number of different parties in various areas of the territory. Which subject and rival parties were chosen for these analyses? Do the data about the rival party refer to the rival party that is closest to the subject's party at the time of data collection? These critical details should be clarified in the main text (not just the methods).

It would be useful to have at least an anecdotal description of whether hilltops provide a better visual vantage point (e.g., less vegetation) or are only thought to enable clearer perception of auditory information.

The article is well written. However, many sentences read as relatively dense because they span multiple lines of text and sometimes include more than two clauses (or a large number of comma-separated units of text). The manuscript's readability could be improved in such cases by breaking these segments into multiple sentences.

Line 191 - when reading the main text, it is not immediately clear whether the dependent measure is all instances where chimpanzees were in the vicinity of a peripheral hill and whether (1) or not (0) they stopped, or all instances where chimpanzees climbed a peripheral hill and whether (1) or not (0) they stopped, or something else. Since the casual reader will focus on the main text, it would be helpful to make this point clearer within that part of the manuscript. I believe it is the second of these options; slight changes in wording would greatly improve clarity (e.g., changing 'all occurrences of peripheral hill use' to 'all instances in which chimpanzees climbed a peripheral hill' AND 'patterns of stop events' to 'the presence or absence of stop events').

Line 291 - if the metric is distance of advance, why is the header speed of advance?

All data appear to be present but would be enriched by inclusion of code (e.g., R scripts) to replicate the authors' analyses

Reviewer #4:

In this paper, Lemoine and colleagues explore the way in which chimpanzees at Tai National Park changed their behavior after resting for 5 min or more on the exposed hilltops in the borders of their territory, especially those borders that overlapped with other groups' territories, with the assumption that their behavior changed due to information gained about the location and size of parties of chimpanzees from other groups. I am enthusiastic about this paper, as I appreciate both the use of a long term data set to answer a really difficult question and their interest in relating it to an important question in human evolution. On the whole, I think that with some edits it will be a good fit in PLOS Biology. That being said, I have some concerns about the analyses and writing, which I address below.

One challenge I had with the paper was figuring out exactly what the authors did. The paper is extremely dense with predictions and analyses and repetitive in places, which made it difficult to follow; I am not certain that I understood their analyses. However, it appears to me that the major findings were based on an analysis of the way chimpanzees behaved when they were on peripheral vs core hills. But to me, a key factor is how they respond when they are on hills vs NOT on hills. Of course, I may have misunderstood, so I have outlined my understanding of the results below, so if I misunderstood the authors can use it to know how to clarify and otherwise they can use it as a suggestion for additional analyses. Looking at each section, I see that:

* Chimps stopped more in hills traveling towards the border than towards the core of their territory, and they were less likely to stop on peripheral hills if they were going to be on peripheral hills later in the day (or maybe they were more likely to stop if they hadn't earlier? Not sure if they can discern causality from this) or if it was later in the day.

* Resting and travel were more likely at high elevation near the border than near the center, which contrasts with feeding, which was more likely at the center. The latter could be that they were more relaxed at the center, but this contrasts with resting, which they shouldn't do if they aren't relaxed unless there is another purpose. However, the graphs in Fig 3 look like the effects are rather modest, so I would like to see some effect sizes on these results. I would also like to know how often they rested on hills in the periphery vs lower areas in the periphery. If they are resting to listen equally often anywhere in the border then they may be using information about rivals tactically, but they aren't using hills tactically, which is different.

* Movements towards rivals' locations were influenced by whether they were heading to the border in the first place, own party size, and rivals' movement direction, plus and interaction between the latter two. I particularly like the quantification of the chimps' risk assessments here (lines 271-4), but I didn't entirely understand what was going on in lines 273-8. It sounds like the chimps counterintuitively were more likely to approach if it was a larger rival group heading towards them? The authors appear to say that this was because the chimps were confused or the rivals were silent (lines 277-8), but wouldn't that be the entire point of using the hills for tactical advantage? 

* Chimps advanced further towards rivals when they were moving towards the border than the core, which is expected as the rivals are presumably at the border. They were more likely to advance if the rivals were a small, close group and they were large, and more likely to retreat if there was a large group of rivals. To be honest, it was not entirely clear how this section (291-305) was different than the previous one (255-278). That could use some more explanation.

* The last section does compare the use of hills vs low elevation areas, and it may be that the information that I have been looking for is here. If so, then I suggest that this be put in earlier models, or addressed first, as I think it is the critical comparison. Indeed, what this section (307-324) appears to suggest is that chimpanzees did stop and rest in low elevation areas, but there is not a comparison to tell us how frequent this was relative to time spent in low vs high elevation areas; that is, are they resting more per time spent in hills or low areas? For instance, what if they're actually resting more in the low areas if they're spending lots of time in the hills, which could suggest that they're going to the hills for safety and they're more nervous in the low areas, hence they are more likely to rest and listen? I'm not suggesting this is the case, but merely pointing out that without that data, we don't know what they're doing. I would like to know how often they are resting in the hills vs in the low areas. Second, it seems that their behavior in low elevations isn't all that different from that in the hills; they are moving forward more from border than core areas and the likelihood to advance and the distance to advance are relative to the party size, just as in the hills. The one difference appears to be that there is no effect of how large the rival's party is or whether they are advancing or retreating. That is potentially the really interesting bit, although effect sizes so that readers can compare relevance. I suspect that these tables (S5B and S6B) should be in the main text for comparative purposes.

Also, all of these analyses chose > 5 min as the cut off for a rest period. How was that time period chosen? That seems like a fairly artificial cutoff; especially if the chimps are nervous and listening, they may pause more briefly to listen and gain information, and sitting still for that long seems risk. Why not look at a shorter period? I would also like to know whether they show the same behavior changes after time in the hills when they are not resting, suggesting that just being in the hills provides that information. That is, are they hearing more just by virtue of being high, and they don't need to stop and rest? Or are they hearing more by virtue of being close to the border, so they don't need to rest? Obviously I don't know what the authors' data structure is, so it may not be possible, but I think this would be very informative.

Finally, these data are highly consistent with what we know of chimpanzee patrols already (they pay attention to their party size and that of their neighbors, and only advance if their party size is the greater), so it would be good to expand upon how what the authors are seeing of their behavior in the hills adds to or changes the understanding we already have.

A second concern is that the authors generally treat the changes in the chimps' actions as if they were actively making a decision to seek out information and change their behavior according to the distribution of neighbors. It wouldn't surprise me if chimpanzees were doing that, but it also wouldn't surprise me if they were not; perhaps they have simply learned to appreciate being high up because they are less likely to get caught unawares, without any understanding of why that is the case? It need not have anything to do with higher order cognition. The authors do mention this at the end of the Discussion, but rather dismissively "Although we make no claims here about social cognition used by the chimpanzees… (line 494)", especially since the entire time the chimps' actions have been referred to as "tactical," which is explicitly suggesting higher order cognition. This possibility needs to be addressed more directly and with more acknowledgement of the fact that we could be seeing this pattern of behavior without any tactics involved.

In general, there are a lot of assumptions that seem as if the authors decided the chimpanzees were behaving tactically and set out to prove it, rather than seeking what the animals did. For instance, in the abstract, the authors say that "A problem for this assertion [that tactical warfare is a driver of human evolution] is that tests in other species are absent." Why is this a problem? It could have emerged after the hominid lineage split from other apes and driven rapid cognitive evolution. Again, I happen to agree with the authors that there is likely evidence in other species, but it is not a given that this statement is correct. Then in lines 83-5 the authors introduce the hypothesis as if it were a fact and only clarify in the next paragraph that it is their hypothesis.

Finally, a smaller point; if elevation is already used routinely for predator detection and signaling, why would the use of elevation in an in-group context suggest tactical usage when these other contexts do not (lines 70-2, 462-474)? Logically, if one suggests higher order cognition, wouldn't they all? This needs to be further addressed in the manuscript.

I have a few general points on the writing as well. The authors tend to write with lots of clauses set off with commas that can make it extremely difficult to follow the main line of argument (the first paragraph is a great example of this). I also really like the idea of Table 1, but it needs additional text to help explain the predictions. A reader should be able to understand it without having to read the rest of the text. Lastly, there is a lot of repetition, particularly in the Discussion, which is quite a bit lengthier than it needs to be. The paper in general would be more impactful if it were more concise.

---

## [Decision Letter · Decision Letter 2]

27 Aug 2023

Dear Dr Lemoine,

Thank you for your patience while we considered your revised manuscript entitled "Tactical use of high elevation in territorial contexts in chimpanzees." for publication as a Research Article at PLOS Biology. This revised version of your manuscript has been evaluated by the PLOS Biology editors, the Academic Editor and one of the original reviewers.

The review is attached below. Based on these comments, we are likely to accept this manuscript for publication, provided you satisfactorily address the remaining points raised by the reviewer and also the data and other policy-related requests stated below.

In addition, we have two editorial requests:

a) Please flip your Title around to include an active verb. We suggest something like "Chimpanzees make tactical use of high elevation in territorial contexts."

b) Reviewer #4 asks you to shorten the paper somewhat, to reduce repetition. While our journal does not have length restrictions, we worry that the length and repetition might compromise the appeal and accessibility of the paper, and we urge you to attend to this issue (as well as the other concerns raised by this reviewer).

We expect to receive your revised manuscript within two weeks. 

*Published Peer Review History*

*Press*

Sincerely,

Ines

--

Ines Alvarez-Garcia, PhD

Senior Editor,

PLOS Biology

on behalf of

Roland Roberts, PhD

Senior Editor

PLOS Biology

rroberts@plos.org

Fig. 3A, B; Fig. 4A-F; Fig. 5A, B; Fig. S2A, B; Fig. S3A, B and Fig. S4

Reviewers' comments

Reviewer #4:

This is a nice revision and the authors have done an excellent job of taking into account the reviewers' earlier comments and concerns. The manuscript is far easier to follow, which also makes it a stronger paper. I have just a few general comments and the rest are editorial points that will help to further clarify the text. I think that this paper is a very nice addition to the literature.

The paper is still quite long and repetitive, so if there are places in which the authors can reduce the text it would be helpful. In particular, there is a lot of overlap between the Results and Discussion. On the one hand, there is a lot going on in this paper, and the additional information is very useful, but if it is possible to reduce repetition, I would do so. Second, I really like the new sections on cognition and implications for human evolution, but I would try to reduce them a bit as they are an interesting implication, rather than the main point. That being said, if the rest of the Discussion were abbreviated I might feel differently! I don't have any specific suggestions on what to cut because I think that at this stage any decisions should be up to the authors and Editor.

Lines 20-23 - this sentence is still very dense and doesn't make a whole lot of sense to someone who hasn't read the paper. Given that this is a general interest journal and I certainly hope this paper is read by those outside of the field (it should be), please try to simplify the sentences.

Line 59 - no comma after power

Line 64 - decision is the wrong word (decision-making?)

Lines 67-72 - I would combine this paragraph with the above. This paragraph is a little short to stand alone.

Line 94 - you don't need to say high elevation twice

Line 194 - less should be fewer (as hills are discrete objects)

Line 202 - neither should be nor

Lines 320-1 - I think you mean that they were equally likely to advance from low elevations (not equally well assessed)?

Line 355 - the sentence starting with "given that" doesn't seem to be complete.

Line 477 - decision-making

---

## [Editor Report · Decision Letter 3]

26 Sep 2023

Dear Dr Lemoine,

Thank you for the submission of your revised Research Article "Chimpanzees make tactical use of high elevation in territorial contexts." for publication in PLOS Biology. On behalf of my colleagues and the Academic Editor, Sarah Brosnan, I'm pleased to say that we can in principle accept your manuscript for publication, provided you address any remaining formatting and reporting issues. These will be detailed in an email you should receive within 2-3 business days from our colleagues in the journal operations team; no action is required from you until then. Please note that we will not be able to formally accept your manuscript and schedule it for publication until you have completed any requested changes.

IMPORTANT: Many thanks for providing the raw data and code as supplementary files. However, we also need the numerical values displayed in Figs 3AB, 4ABCDEF, 5AB, S2AB, S3AB, S4. In additional, please cite the location of the data clearly in the each relevant main and supplementary Figure legend, e.g. "The data underlying this Figure may be found in S1 Data" - I have asked my colleagues to include this request alongside their other format-related requirements.

Sincerely, 

Roli Roberts

Senior Editor

PLOS Biology

rroberts@plos.org